# Photoreactivity of Hair Melanin from Different Skin Phototypes—Contribution of Melanin Subunits to the Pigments Photoreactive Properties

**DOI:** 10.3390/ijms22094465

**Published:** 2021-04-24

**Authors:** Krystian Mokrzynski, Shosuke Ito, Kazumasa Wakamatsu, Theodore G. Camenish, Tadeusz Sarna, Michal Sarna

**Affiliations:** 1Department of Biophysics, Faculty of Biochemistry, Biophysics and Biotechnology, Jagiellonian University, 30-387 Krakow, Poland; mokrzynskikrystian@gmail.com (K.M.); tadeusz.sarna@uj.edu.pl (T.S.); 2Institute for Melanin Chemistry, Fujita Health University, Toyoake, Aichi 470-1192, Japan; sito@fujita-hu.ac.jp (S.I.); kwaka@fujita-hu.ac.jp (K.W.); 3Department of Biophysics, Medical College of Wisconsin, Milwaukee, WI 53226, USA; tgcam@mcw.edu

**Keywords:** melanoma susceptibility, skin phototypes, melanin pigmentation, eumelanin, pheomel-anin, DHI-melanin, DHICA-melanin, BZ-melanin, BT-melanin, photoreactivity, singlet oxygen

## Abstract

Photoreactivity of melanin has become a major focus of research due to the postulated involvement of the pigment in UVA-induced melanoma. However, most of the hitherto studies were carried out using synthetic melanin models. Thus, photoreactivity of natural melanins is yet to be systematically analyzed. Here, we examined the photoreactive properties of natural melanins isolated from hair samples obtained from donors of different skin phototypes (I, II, III, and V). X-band and W-band electron paramagnetic resonance (EPR) spectroscopy was used to examine the paramagnetic properties of the pigments. Alkaline hydrogen peroxide degradation and hydroiodic acid hydrolysis were used to determine the chemical composition of the melanins. EPR oximetry and spin trapping were used to examine the oxygen photoconsumption and photo-induced formation of superoxide anion, and time-resolved near infrared phosphorescence was employed to determine the singlet oxygen photogeneration by the melanins. The efficiency of superoxide and singlet oxygen photogeneration was related to the chemical composition of the studied melanins. Melanins from blond and chestnut hair (phototypes II and III) exhibited highest photoreactivity of all examined pigments. Moreover, melanins of these phototypes showed highest quantum efficiency of singlet oxygen photogeneration at 332 nm and 365 nm supporting the postulate of the pigment contribution in UVA-induced melanoma.

## 1. Introduction

Melanin pigment is commonly viewed as a natural protectant against harmful effects of solar radiation [1]. Although the modulatory effect of melanin in melanoma aggressiveness has been demonstrated [2,3], it remains unclear whether the pigment plays a role in melanomagenesis. Due to the postulated involvement of the pigment in UVA-induced melanoma [4] the photoreactivity of melanin has recently become a major focus of research. Until now it was commonly viewed that the increased susceptibility of light skin individuals to UVA-induced melanoma has been associated with low levels of their skin pigmentation [5]. Thus, individuals of phototype I, which contain lowest amounts of the pigment were considered as being least protected from harmful effects of solar radiation [6]. However, recent data indicate that individuals of phototype II and III, are at least as susceptible to UVA-induced melanoma as those of phototype I [7]. Importantly, the skin of individuals with phototype II and III is not only more pigmented than the skin of individuals with phototype I, but it also contains more eumelanin, which is commonly viewed as a photoprotective pigment [8]. These unexpected observations corroborated a hypothesis about the role of melanin photoreactivity in melanomagenesis [9,10]. Indeed, it was recently demonstrated that photoexcitation of melanin resulted in generation of reactive oxygen species (ROS), in particular singlet oxygen [11]. Moreover, it was shown that singlet oxygen generated by melanin could induce characteristic oxidative DNA damage (8-oxo-7,8-dihydroguanine), leading to mutation [12,13]. It is important to realize that the generation of singlet oxygen requires the formation of long-lived triplet excited state of melanin [14,15]. Independent studies demonstrated that this key reactive intermediate of melanin was indeed generated either via photoexcitation or chemical reactions [16,17]. However, different melanin pigments may exhibit different photochemical properties. Thus, while eumelanin is viewed as a good photoprotective agent [18] that exhibits only residual photoreactivity, pheomelanin is believed to be more photoreactive and can act even acting as a potent photosensitizer [19]. Importantly, natural melanins are almost always a mixture of both eumelanin and pheomelanin at different ratios [20,21]. While eumelanin is predominantly composed of 5,6-dihydroxyindole (DHI) and 5,6-dihydroxyindole-2-carboxylic acid (DHICA), the main pheomelanin subunits are mostly benzothiazine (BT) and benzothiazole (BZ) moieties [22,23]. Depending on the pigment origin and exposure to environmental factors, the ratio of these subunits may vary significantly [24]. Although it is unclear how photoreactivity of melanin pigments is exactly determined by chemical nature of the melanin subunits, results of our previous studies demonstrated a distinct relationship between the photoreactivity of synthetic melanins and their chemical composition [25,26]. Thus, it was reported that synthetic 5-*S*-cysteinyldopa melanin, containing high percentage of BT subunits, was significantly less photoreactive than the photodegraded melanin, which contained higher content of modified BZ moieties. The proposed mechanism of UVA-induced oxidation of DHICA melanin involved both superoxide anion and singlet oxygen [27], and in a follow-up study it was demonstrated that DHICA melanin photogenerated singlet oxygen with the efficiency almost three-fold higher than DOPA-melanin and reacted with singlet oxygen at least five-fold faster than other synthetic melanins [11]. Taking into consideration the postulated role of melanin photoreactivity in melanomagenesis, the ability of natural melanins to photogenerate reactive oxygen species should be thoroughly examined.

In this work, melanin isolated from hair samples obtained from donors of different skin phototypes was examined for its photoreactive properties. The obtained results showed that melanin from blond and chestnut hair (phototypes II and III) exhibited highest photoreactivity of all examined pigments, especially in the UVA region.

## 2. Results

Hair samples were collected from donors of different skin phototypes (three white individuals from Europe (phototypes I-III) and one black individual from Africa (phototype V). Characteristic features of the donors were as following: phototype I (white male, red hair, blue eyes, pale skin), phototype II (white female, blond hair, green eyes, light skin), phototype III (white female, chestnut hair, brown eyes, olive skin), and phototype V (black male, black hair, dark-brown eyes, dark skin). All individuals were of similar age, non-smokers and did not use any hair dye or colorizing shampoo. Figure 1 shows photographs of hair samples obtained from the donors before isolation of the pigments. 

During isolation of the melanins two fractions were obtained. These were: spin-down melanosomes and melanin nanoaggregates in the supernatant, which broke off from the pigment granules during isolation (Figure 2). Both fractions were used in the analysis.

### 2.1. Paramagnetic Properties of the Isolated Melanins

To determine the paramagnetic properties of the melanins, electron paramagnetic resonance (EPR) spectroscopy was employed [28]. Figure 3A–C shows the X-band EPR spectra of melanosomes, melanin nanoaggregates, and synthetic melanin models, respectively. As evident, red hair melanin had a signal similar to that of Cys-L-Dopa melanin used as standard of pheomelanin in that the lower field component was visible, whereas EPR spectra of blond, chestnut, and black hair melanins resembled that of DOPA melanin used as standard of eumelanin with only one spectral line visible. Although X-band EPR spectroscopy is routinely used to examine melanins of different origin, it is relatively low spectral resolution does not allow to detect low levels of pheomelanin in mixed melanin samples. Thus, to reduce the ambiguity of melanin identification associated with X-band EPR, we employed W-band EPR, which operating at ten-fold higher microwave frequency provided much higher spectral resolution than X-band EPR [25]. 

Figure 3D–F shows the W-band EPR spectra of melanosomes, melanin nanoaggregates and synthetic melanin models, respectively. As evident from W-band EPR measurements, black hair melanosomes and melanin nanoaggregates strongly resemble the spectrum of synthetic eumelanin. In the case of red hair, only melanosomes exhibited EPR signal comparable to the EPR spectrum of synthetic pheomelanin. Interestingly, the EPR signal of red hair nanoaggregates more resembled that of modified pheomelanin, such as photodegraded synthetic 5-*S*-cysteinyldopa melanin [25]. EPR spectra of blond hair samples exhibited some features that can also be found in 5-*S*-cysteinyldopa melanin, most notably the overall spectral width of the signal. However, certain characteristics of the signals, such as the pronounced maximum at *g* = 2.0025 are more consistent with partially degraded pheomelanin, as observed in photobleached 5-*S*-cysteinyldopa melanin [25]. This is particularly evident for blond hair melanin nanoaggregates. W-band EPR spectra of chestnut hair melanosomes were substantially distorted by superposition of the third hyperfine line of manganese (II) that overlapped with the main signal and the fourth line visible at *g* below 2. Nevertheless, considering the overall spectral width of the W-band EPR signals of chestnut melanosomes and melanin nanoaggregates, we can conclude that the predominant type of the sample melanin was modified eumelanin. Although the central part of the EPR spectrum of melanin nanoaggregates from chestnut hair is significantly modified compared to EPR spectra of DOPA-melanin and samples from black hair, the low-field and high-field extrema of the chestnut hair nanoaggregates are consistent of eumelanin.

### 2.2. Chemical Analysis of the Isolated Pigments

To determine the chemical composition of the pigments, alkaline hydrogen peroxide oxidation (AHPO) and hydroiodic acid (HI) hydrolysis were employed. Ratios of selected markers of chemical degradation of the isolated pigments are shown in Figure 4 whereas the exact values of the analyzed markers are shown in Appendix A. 

Hence the exact amount of melanin could not be accurately measured due to low quantity of the pigment in the samples, results are presented by normalization to the absorption at 500 nm (A500). The A500 value obtained by Soluene-350 solubilization reflects a total amount of melanin [29]. The A650/A500 ratio indicates whether melanin is eumelanic (ratio > 0.20) or pheomelanic (< 0.15). AHPO yields pyrrole-2,3,5-tricarboxylic acid (PTCA), pyrrole-2,3-dicarboxylic acid (PDCA), and thiazole-2,4,5-tricarboxylic acid (TTCA), as markers of DHICA and DHI moieties of eumelanin and benzothiazole (BZ) moiety of pheomelanin, respectively [21]. HI hydrolysis yields 4-amino-3-hydroxyphenylalanine (4-AHP) and 3-amino-4-hydroxyphenylalanine (3-AHP) as markers of 5-*S*-cysteinyldopa- and 2-*S*-cysteinydopa-derived benzothiazine (BT) moieties, respectively [30]. Gross comparison of the results on melanosomes and melanin nanoaggregates indicated that nanoaggregates were more pheomelanic than the corresponding melanosomes, as evidenced by the lower A650/A500 and higher AHPs (4-AHP + 3-AHP)/PTCA ratios, except nanoaggregates from black hair. Red melanosomes and melanin nanoaggregates were typical pheomelanic, as revealed by low A650/A500, low PTCA/A500, high AHPs/A500, and high TTCA/A500 ratios [29,31]. Blond hair melanosomes and nanoaggregates were mixed-type melanin, as suggested by low to medium A650/A500 ratios. It should be stressed that blond hair melanosomes and melanin nanoaggregates showed higher TTCA/4-AHP (an indicator of BZ/BT ratio) ratios than the red hair counterparts, indicating photoaging of pheomelanin in blond hair melanins, as have been shown in photoexposed natural and synthetic pheomelanins [24,32]. This photoaging is consistent with the lower 4-AHP/3-AHP, AHPs/A500, and AHPs/PTCA ratios in blond hair melanosomes and melanin nanoaggregates than those in red hair [24,32]. Both black and chestnut hair melanosomes and melanin nanoaggregates were typical eumelanic, as revealed by high A650/A500, high PTCA/A500, trace 4-AHP/A500, and low TTCA/A500 ratios [29,31]. However, the A650/A500 ratios in chestnut hair melanin nanoaggregates were much lower than those in black hair, suggesting contribution from pheomelanin, as supported by the higher AHPs/A500 ratio [29,31]. The most striking difference between melanosomes and melanin nanoaggregates from chestnut and black hair was found in the PTCA/PDCA ratio (an indicator of DHICA/DHI ratio) [33]. Melanosomes in chestnut and black hair showed much higher PTCA/PDCA (DHICA/DHI) ratios than the corresponding nanoaggregates, suggesting higher susceptibility of DHICA units of melanin nanoaggregates to undergo degradation. 

### 2.3. EPR Analysis of Oxygen Photoconsumption and Photogeneration of Superoxide Anion

Irradiation of suspensions of melanosomes or solutions of melanin nanoaggregates with 365 nm (Figure 5A,B) and with 445 nm (Figure 5D,E), resulted in a steady drop in the concentration of dissolved oxygen. For melanosomes irradiated with either 365 or 445 nm, the highest rate of oxygen consumption was observed in the case of chestnut hair followed by blond hair and red hair with the black hair having the slowest consumption rates. Interestingly, melanin nanoaggregates exhibited different photoreactivity than the corresponding melanosomes. When irradiated at 365 or 445 nm, melanin nanoaggregates from blond hair consumed oxygen most efficiently with the corresponding rates being almost two times higher than that of nanoaggregates from chestnut hair. These rates of photoconsumption of oxygen were one-and-half to two-orders of magnitude higher than the rates observed for the corresponding melanosomes. Melanin nanoaggregates from red hair and black hair photoconsumed oxygen significantly less efficiently with their rates being comparable to those of the corresponding melanosomes. Calculated values of the initial rates of oxygen photoconsumption are shown in column graphs in Figure 5C,F whereas the exact numerical values can be found in Appendix A. The 365 nm light is about five-fold more efficient than 445 nm light in inducing oxygen photoconsumption in samples containing hair melanosomes. Irradiation of melanin nanoaggregates with light causes more diverse effects. Thus the UVA, in comparison with the short wavelength visible light, increases the rate of oxygen photoconsumption by four to ten times, depending on the type of melanin nanoaggregates. 

Kinetics of superoxide anion formation, as a result of the interaction of melanin with dioxygen induced by irradiation with UVA or violet light are shown in Figure 6A,B and Figure 6D,E respectively. Calculated initial rates of the DMPO-OOH adduct generation are shown in column graphs in Figure 6C,F whereas the exact numerical values can be found in Appendix A. Initial growth rate of the corresponding EPR signal, which can be attributed to the efficiency of the pigments to photogenerate superoxide anion, was significantly higher in samples containing melanin nanoaggregates than melanosomes. Black hair melanin samples photogenerated superoxide anion with the lowest rate, followed by red hair samples. UVA, in comparison with violet light, stimulated oxygen photoconsumption 3–9-fold more efficiently in case of melanosomes and 5–23 times faster in case of melanin nanoaggregates. The photogeneration of superoxide anion mediated by melanin nanoaggregates irradiated at 445 nm was 5 to 29 times faster than that mediated by the corresponding melanosomes. The effect was even more diverse in case of melanin samples irradiated at 365 nm. Thus, while melanin nanoaggregates from red and black hair were 6–7 times more efficient than their corresponding melanosomes to photogenerate superoxide anion, melanin nonoaggregates from blond hair were 85 times more efficient. Highest initial rates of DMPO-OOH adduct formation, during sample irradiation with either 365 or 445 nm, was observed for nanoaggregates derived from blond hair. Changes in the intensity of the EPR signal of DMPO-OOH in this sample demonstrated biphasic character—a rapid accumulation of the signal (a maximum reached by approximately 120 s), proceeded its slower decay with a minimum reached after about 600 s. The kinetics of the DMPO-OOH signal can be compared with the kinetics of oxygen photoconsumption. It is apparent that the spin adduct signal begins to decay when oxygen is substantially depleted from the sample. Thus it can be inferred that melanin-reducing equivalents activated by UVA are responsible for the reduction of the spin adduct to its diamagnetic form.

### 2.4. Time-Resolved Singlet Oxygen Phosphorescence Detection

To examine the ability of the hair melanins to photogenerate singlet oxygen, time-resolved near-infrared luminescence, was employed. Prior to the measurements, UV-Vis absorption spectra of melanin nanoaggregates were obtained (Figure 7A). Normalized absorption spectra revealed that blond hair melanin had the lowest absorbance of all examined pigments in the spectral range 300–600 nm. Although melanin nanoaggregates from red and chestnut hair exhibited in the visible part of the spectrum lower absorption compared to nanoaggregates from black hair melanin, the relationship was reversed in the UVA region. Action spectra of singlet oxygen photogeneration by the examined melanin nanoaggregates are shown in Figure 7B. To confirm the singlet oxygen origin of the obtained phosphorescence signals, additional experiments, including the effect of oxygen for argon exchange and the effect of physical quenchers on measured phosphorescence signal were performed (data not shown). The obtained data show a significant increase in photogeneration of singlet oxygen with decreasing wavelength, particularly between 420 and 350 nm. 

To determine if the observed properties of melanin nanoaggregates to photogenerate singlet oxygen could be assigned to specific monomer units found in eumelanin and pheomelanin, the generation of singlet oxygen by selected model compounds, were also tested. Figure 8A shows absorption spectra of: DHBTCA [7-(2-amino-2-carboxyethyl)-5-hydroxy-3,4-dihydro-2*H*-1,4-benzothiazine-3-carboxylic acid], BZ-AA [6-(2-amino-2-carboxyethyl)-4-hydroxybenzothiazole], DHICA, and DHI, whereas Figure 8B shows action spectra of singlet oxygen generation by the monomers. Comparison of the figures indicates that the spectral dependence of photogeneration of singlet oxygen by the model melanin subunits generally follows their absorption.

The collective Figure 9 shows superposition of the absorption spectra and action spectra of singlet oxygen generation for selected melanin nanoaggregates, and for the compounds proposed as key melanin subunits. It is apparent that the obtained action spectra of singlet oxygen photogeneration by melanin nanoaggregates differ from their absorption spectra. Partial similarity of both spectra, can only be found for nanoaggregates from red and, to a less extent, black hair melanins. For nanoaggregates from blond and chestnut hair, the greatest discrepancy between the absorption and action spectra is evident in the 435–350 nm region, where the efficiency of melanin nanoaggregates to photogenerate singlet oxygen is significantly higher than their efficiency to absorb light. The figure also indicates that the model compounds in the form used by us cannot be responsible for the observed photogeneration of singlet oxygen by melanin nanoaggregates in the spectral region above 350 nm.

Although the determined action spectra for photogeneration of singlet oxygen give a reasonable qualitative characterization of the melanin ability to generate this reactive oxygen species at different excitation wavelength, quantum yield of singlet oxygen photogeneration provides quantitative information about melanin photoreactivity [11]. This important photophysical parameter of melanin nanoaggregates was determined by a comparative method using two different standards applicable for the visible and UV part of the spectrum. One was proflavine, reported to photogenerate at 365 nm singlet oxygen with the yield 0.12, and the other was fluorescein, with the reported yield to photogenerate singlet oxygen at 445 nm with the yield 0.03 to 0.06 [34,35]. First, quantum yields for photogeneration of singlet oxygen by the two standards were verified, and the corresponding values were found: 0.11 for proflavine and 0.034 for fluoresceine. Results for melanin nanoaggregates are shown in Figure 10, whereas numerical values are present in Table 1. The determined quantum yields significantly differed between melanin nanoaggregates of different hair. The highest values were obtained for nanoaggregates from blond hair followed by nanoaggregates from chestnut hair, with the lowest values observed for nanoaggregates from black hair and red hair. For blond hair and chestnut hair melanins their quantum yields were significantly higher in the UV region than in the visible region with the highest quantum yields for these two melanin nanoaggregates determined at 332 nm. At shorter wavelengths (300 nm) the values were actually lower. The relatively high quantum yields for singlet oxygen generation by nanoaggregates from blond and chestnut hair in the UVA (3–8%) are of considerable interest suggesting high photosensitizing potential of these melanins. The ability of natural melanins to quench singlet oxygen was also addressed in our study and the obtained data are summarized in Table 1. It is apparent that all melanin nanoaggregates quenched singlet oxygen with a similar rate constant—1.1–2.2×10^5^ (mg/ml) ^−1^s^−1^. The determined quenching rate constants are of the same order of magnitude as those obtained for synthetic melanins [11].

Using similar experimental approach, quantum yields for singlet oxygen photogeneration by the selected model compounds were determined (Appendix A), whereas numerical values are shown in Table 2. Surprisingly these values are lower than the corresponding values obtained for the most photoreactive melanin nanoaggregates. 

## 3. Discussion

The obtained results clearly demonstrated that natural melanins isolated from human hair exhibited differential photoreactivity, which depended on the excitation wavelength and physicochemical form of the isolated pigments. This issue is best illustrated by the ability of melanosomes and melanin aggregates to mediate consumption of oxygen induced by violet and UVA light (Figure 5). Although oxygen consumption does not provide specific information about the exact mechanism responsible for the examined process, it is very useful for monitoring progress of aerobic photochemical processes [16]. The consumption of oxygen mediated by hair melanosomes increased five to six-fold after changing the irradiation wavelength from 445 nm to 365 nm, indicating, not surprisingly, higher efficiency of short-wavelength light to activate melanin. It is not clear why the effects are stronger for melanin nanoaggregates from blond and chestnut hair (10–11 fold) and weaker for nanoaggregates from red and black hair (about four-fold). It is important to realize that oxygen photoconsumption depends on the formation of reactive oxygen species and their interaction with appropriate substrates. In the samples studies, both were provided by the melanins. It is likely that different chromophores in the melanin nanoaggregates from blond and chestnut hair, compared to those from red and black hair were involved in the studied photoprocesses. The chromophores of the nanoaggregates from blond and chestnut hair were presumably not only more photoreactive, they also exhibited stronger wavelength dependence of their photoactivation.

An intriguing issue is the dramatic increase in the rates of photoconsumption of oxygen observed for nanoaggregates from blond and chestnut hair compared to their corresponding melanosomes. Although in part, it could be attributed to significantly smaller size of the melanin nanoaggregates in comparison with the corresponding melanosomes (different by at least an order of magnitude), making the surface to volume ratio substantially higher for the nanoaggregates, thus increasing the accessibility of their functional groups to external agents, the fact that melanosomes and melanin nanoaggregates from red and black hair did not follow the same dependence, suggesting that other factors also played a role. The rates of photoconsumption of oxygen were generally higher for melanosomes from blond and chestnut hair, compared to black and red hair. It suggests that melanosomes from blond and chestnut hair contain chromophores with high photochemical activity. An increased exposure of these photochemically active groups to the environment, facilitated by the melanosome fragmentation, would result in a significant enhancement of the observed photoreactivity such as elevated oxygen photoconsumption and release of reactive oxygen species. 

One of the reactive oxygen species photogenerated by melanin is superoxide anion—product of one electron reduction of molecular oxygen by melanin reducing equivalents. Different efficiencies of melanin samples examined in this study to generate superoxide anion upon irradiation with violet or UVA light are shown in Figure 6. Consistent with oxygen photoconsumption, melanin nanoaggregates from blond and chestnut hair exhibited by far the highest efficiency to generate superoxide anion, particularly when excited by UVA. However, the difference in generation of superoxide anion by different melanin nanoaggregates is not as great as that observed in photoconsumption of oxygen. While black hair melanin nanoaggregates photogenerated superoxide anion at 365 nm and 445 nm, about twenty- and ten-fold fold slower than nanoaggregates from blond hair, the corresponding rates of oxygen photoconsumption were 250 and over 100 fold higher. It may indicate that other reactive oxygen species, such as singlet oxygen, contribute to the observed oxygen photoconsumption. Indeed the results presented in the previous chapter show that melanin nanoaggregates generated singlet oxygen with relatively high efficiency that depended on the excitation wavelength much weaker that photogeneration of superoxide anion.

It is rather remarkable that quantum yield for singlet oxygen photogeneration, determined at 332 nm, was above 8% for melanin nanoaggregates from blond and 6% for chestnut hair nanoaggregates; even at 365 nm, the corresponding yields were 3–4%. The data suggest that melanin nanoaggregates can act as an efficient photosensitizer generating singlet oxygen. This powerful oxidizing agent is known for its ability to oxidatively damage proteins, nucleic acids, and unsaturated lipids [36].

Detailed characterization of the aerobic photoreactivity of melanosomes and melanin nanoaggregates from human hair of different color revealed that contrary to common belief red hair melanin is not an efficient photosensitizing agent. In the UVA and short wavelength visible light, the ability of red hair melanin to photogenerate superoxide anion and singlet oxygen is relatively low, similar to black hair melanin. Similar conclusion was reached in a study published almost forty years ago [37] and in a more recent study, which compared photogeneration of superoxide anion and singlet oxygen by synthetic eumelanins and pheomelanins [11]. 

The high aerobic photoreactivity of melanin nanoaggregates from blond and chestnut hair is of special interest. Although the exact physicochemical basis for this phenomenon remains unknown, it can be postulated that the presence of modified pheomelanin constituents, particularly in nanoaggregates from blond hair plays a substantial role. Indeed, we have recently demonstrated that aerobic photodegradation of synthetic pheomelanin obtained for 5-*S*-cysteinyldopa is accompanied by significant increase in its ability to photogenerate singlet oxygen [25]. The chemical degradation analysis and W-band EPR spectroscopy measurements seem to be consistent with the assumption that blond hair melanin and to less extent chestnut hair melanin contain modified benzothiazole units, an indication of aerobic photodegradation of the pheomelanin component. Such benzothiazole moieties are expected to have higher aerobic photoreactivity than benzothiazine units. The high aerobic photoreactivity of nanoaggregates from chestnut hair remains a puzzle. Although W-band EPR spectroscopy and chemical degradation analysis indicate that the dominant type of melanin from chestnut hair is eumelanin, we postulate that it is highly modified eumelanin, similar to that obtained by photobleaching of melanosomes from porcine and bovine retinal pigment epithelium [38,39]. 

Even though the aerobic photoreactivity of hair melanin seems of limited biological relevance, except for the pigment photostability, the examined melanins could also be present in the human skin depending on the skin phototype. In particular, melanin similar to the blond hair melanin is expected to be present in the skin of Caucasians of light pigmentation. Such melanin, due to extensive exposure of the skin to solar radiation, could undergo at least partial degradation resulting in elevated photoreactivity of the melanin which exhibits enhanced photogeneration of reactive oxygen species. 

## 4. Materials and Methods

### 4.1. Isolation of Melanin Pigments

Natural melanin pigments were isolated from human hair according to a procedure described previously [40]. First, the obtained hair samples were repeatedly washed with acetone (Sigma-Aldrich, Saint Louis, MO, USA) and sequentially rinsed with dichloromethane and diethyl ether (Sigma-Aldrich, Saint Louis, MO, USA). Subsequently, hair samples were washed with acetone, water, and again acetone and cut into short fragments approximately 2–5 mm long. Obtained hair samples were homogenized in 0.1 M phosphate buffer (Sigma-Aldrich, Saint Louis, MO, USA) (15 ml per 1 g of sample) using porcelain pestle homogenize and transferred into separate flasks. Next, 1,4-dithiothreitol (DTT) (Sigma-Aldrich, Saint Louis, MO, USA) (0.2 g per 1 g of sample) was added to the homogenates and the resulting mixtures were incubated at 37 °C for 18 h. Proteinase K (A&A Biotechnology, Gdansk, Poland) (8 mg per 1 g of sample) and DTT (0.1 g per 1 g of sample) were then added to the mixtures and incubated at 37 °C for additional 18 h. The mixtures were centrifuged for 20 min (3300× *g*, 4 °C). The supernatants were transferred to a separate flask and dialyzed for 72 h using dialysis membrane with water changed every 12 h. Obtained pellets were suspended in water and centrifuged for 10 min (3300× *g*, 4 °C). Resulting residues were suspended in 10 ml of 0.1 M phosphate buffer and solution of 2% *w/v* Triton X-100 (Sigma-Aldrich, Saint Louis, MO, USA) was added up to 1% *w/v* final concentration. The resulting mixtures were stirred for 4 h at room temperature and centrifuged for 20 min (3300× *g*, 4 °C), suspended in water and centrifuged for 10 min (3300× *g*, 4 °C). Finally, the pellets collected by centrifugation, were suspended in 1 ml of ultra-pure water and stored at 4 °C. 

### 4.2. X-Band EPR Spectroscopy

X-band EPR measurements were performed using Bruker EMX-AA EPR spectrometer (Bruker Biospin, Rheinstetten, Germany), according to the method described elsewhere [41]. In brief, aqueous samples of melanosomes and melanin nanoaggregates, saturated with zinc acetate, after freezing in liquid nitrogen, were run at 77 K using a quartz finger-type dewar. EPR spectra were obtained using the following parameters: 32.4 µW microwave power, 0.305 mT modulation amplitude, 336.2 mT center field, 7 mT scan range, 327.7 µs time constant, and 42 s scan time. Ten scans were averaged to obtain the final EPR spectra.

### 4.3. W-Band EPR Spectroscopy

W-band EPR spectra were obtained using a spectrometer system constructed at the National Biomedical EPR Center at Medical College of Wisconsin, USA. Samples in 0.2 mm ID and 0.33 OD synthetic silica capillaries (Fiber Optic Center, New Bedford, MA, USA) were placed in the cylindrical TE011 cavity resonator, specially designed for aqueous samples operating at 94.04 GHz [42]. Both sample and resonator were maintained at 25 °C using the temperature-controlled water bath circulating through a clamp attached to the resonator assembly. The samples were examined in 20 mM acetate buffer using the following parameters: 50 µW microwave power, 937 Hz field modulation frequency, 0.4 mT peak-to-peak modulation amplitude, 20 mT magnetic field scan width. The obtained spectra were the result of multiple scans averaged over a period of time required for adequate signal to noise ratio (20–40 min).

### 4.4. Chemical Analysis of Melanin Subunits

A500 (Absorbance at 500 nm) and A650 values were measured using Soluene-350 solubilization [29]. Pyrrole-2,3-dicarboxylic acid (PDCA), pyrrole-2,3,5-tricarboxylic acid (PTCA), and thiazole-2,4,5-tricarboxylic acid (TTCA), which are the markers of 5,6-dihydroxyindole (DHI), 5,6-dihydroxyindole-2-carboxylic acid (DHICA) eumelanin subunits, native and modified benzothiazole units of pheomelanin, respectively, were measured using alkaline H_2_O_2_ oxidation (AHPO) as described elsewhere [21]. These markers were analyzed by the improved method of HPLC [43]. 4-Amino-3-hydroxyphenylalanine (4-AHP) and 3-amino-4-hydroxyphenylalanine (3-AHP), which are the markers of 5-*S*-cysteinyldopa and 2-*S*-cysteinyldopa, respectively, were measured using hydroiodic acid (HI) hydrolysis as described elsewhere [30]. 

### 4.5. Oxygen Consumption Measurements

Time-dependent changes in oxygen concentration induced by light were determined by electron paramagnetic resonance (EPR) oximetry using mHCTPO (4-protio-3-carbamoyl-2,2,5,5-tetraperdeuteromethyl-3-pyrrolin-1-yloxy) at 0.1 mM concentration as dissolved oxygen-sensitive spin probe, according to the method described elsewhere [44]. In short, samples containing 0.122 mg/ml of melanins in PBS pH = 7.7 were irradiated in EPR quartz flat cell in the resonant cavity using 365 nm light (4.5 mW/cm^2^) or 445 nm light (14 mW/cm^2^) generated by LED light chips (High Power LED Chip, Chanzon, Shen-Zhen, Guangdong, China). EPR oximetry measurements were conducted using the following parameters: 1.06 mW microwave power, 0.006 mT modulation amplitude, 0.3 mT scan width, and 5.2 s scan time, employing the same X-band EPR spectrometer as described above.

### 4.6. EPR Spin Trapping Studies

EPR spin trapping of photogenerated radicals were performed using 100 mM DMPO (5,5-Dimethyl-1-Pyrroline *N*-oxide) (Sigma-Aldrich, Saint Louis, MO, USA) as a spin trap, according to the procedure described elsewhere [45]. In brief, samples containing 0.061 mg/ml melanin in 70% DMSO (Sigma-Aldrich, Saint Louis, MO, USA) were irradiated employing the same light source as described for oxygen photoconsumption. EPR spin trapping was carried out using the following apparatus parameters: 10.6 mW microwave power, 0.05 mT modulation amplitude 339.0 mT center field, 8 mT scan field and 84 s scan time. Simulation of the obtained EPR spectra was performed employing WinSim 2002 software (National Institute of Environmental Health Sciences, Research Triangle Park, Durham, NC, USA). 

### 4.7. UV-Vis Spectroscopy

The optical absorption of melanin nanoaggregates, native DHI and DHICA (subunits of eumelanin) monomers, and DHBTCA and BZ-AA (subunits of pheomelanin) were measured in D_2_O (Sigma-Aldrich, Saint Louis, MO, USA) using HP 8453 diode array spectrophotometer (Hewlett-Packard, Palo Alto, CA, USA). The obtained spectra were corrected by the subtraction of the value at 600 nm from the values at lower wavelengths and normalized in the range 300–600 nm.

### 4.8. Time-Resolved Singlet Oxygen Phosphorescence

Formation and decay of singlet oxygen in samples of natural melanins, eumelanin subunits (DHI, DHICA) and pheomelanin subunits (DHBTCA, BZ-AA) dissolved in phosphate-buffered D_2_O (pH 7) was examined according to the procedure described elsewhere [46]. Hence, samples in a 10-mm optical path quartz fluorescence cuvette (QA-1000; Hellma, Mullheim, Germany) were excited with light pulses generated by an integrated nanosecond DSS Nd:YAG laser system equipped with a narrow-bandwidth optical parameter oscillator (NT242-1k-SH/SFG; Ekspla, Vilnius, Lithuania). Photoexcited generation of singlet oxygen by melanin nanoaggregates was examined in the spectral range 300–600 nm. To determine the quantum yield of singlet oxygen photo-generation by the examined samples, proflavine (Sigma-Aldrich, Saint Louis, MO, USA) and fluorescein (Sigma-Aldrich, Saint Louis, MO, USA) were used as a reference for 365 nm and 330nm, 332 nm, 445 nm, respectively. The quantum yield of singlet oxygen photogeneration by the reference dyes was determined by comparison with that of Rose Bengal (Sigma-Aldrich, Saint Louis, MO, USA) used as a standard. Quantum yields of singlet oxygen generation was determined by comparative measurements of the initial intensities of 1270 nm phosphorescence emitted by the examined samples and reference dyes excited with laser pulses of increasing energies. Melanins and dyes were dissolved in D_2_O and their absorbance was adjusted to 0.123 ± 0.001, 0.116 ± 0.005, 0.140 ± 0.003 and 0.105 ± 0.004 at 300 nm, 332 nm, 365 nm and 445 nm, respectively. Rose Bengal dissolved in D_2_O (absorbance at 550 nm ~0.15), excited with 550-nm laser pulses, was used as a photo-sensitizer in experiments designed to determine rate constants of singlet oxygen quenching by natural melanins. To adjust photoexcitation energy in experiments designed to determine quantum yield of singlet oxygen generation and quenching of singlet oxygen, laser beam was attenuated with 2–4 pieces of wire mesh (light transmission of each piece ~30%). All samples were constantly mixed during measurement using a dedicated magnetic stirrer. The near-infrared luminescence was measured perpendicularly to the excitation beam in a photon-counting mode using a thermoelectric cooled NIR PMT module (H10330-45; Hamamatsu, Japan) equipped with a 1100-nm cut-off filter with an additional dichroic narrow-band filter NBP, selectable from the spectral range 1150–1355 nm (NDC Infrared Engineering Ltd, Bates Road, Maldon, Essex, UK). Data were collected using a computer-mounted PCI-board multichannel scaler (NanoHarp 250, PicoQuant GmbH, Berlin, Germany). Data analysis, including first-order luminescence decay fitted by the Levenberg-Marquardt algorithm was performed using custom-written software. Acquisition time for obtaining melanin action spectra was 40 s for each examined wavelength.

### 4.9. Statistical Analysis

All experiments were performed at least three times. Statistical analysis of the data was made using OriginPro software (OriginLab, Northampton, MA, USA).

## 5. Conclusions

This study systematically analyzed photoreactive properties of natural melanins from hair samples obtained from volunteers of different skin phototypes. The efficiency of the examined melanins to generate superoxide anion and singlet oxygen significantly increased with decreasing excitation wavelength, particularly in the UVA part of the spectrum. Melanin nanoaggregates from blond and chestnut hair, photoexcited with long-wavelength UV were found to generate singlet oxygen with remarkably high efficiency. Based on chemical degradation analysis and W-band EPR spectroscopy measurements it is postulated that the high aerobic photoreactivity of blond and chestnut hair melanins can be attributed to modified pheomelanin and eumelanin components of these pigments. Presence of such modified melanins in the human skin could elevate the risk of phototoxic reactions involved in UVA-induced melanoma.

## Figures and Tables

**Figure 1 ijms-22-04465-f001:**
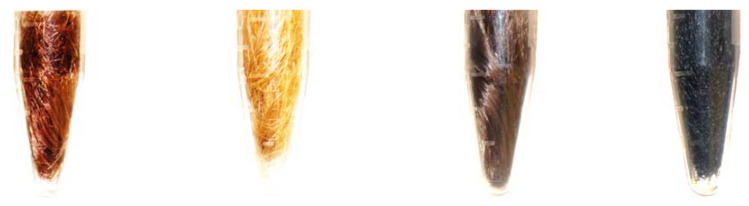
Images of hair samples obtained from donors of different skin phototypes (from **left** to **right**): red hair (phototype I), blond hair (phototype II), chestnut hair (phototype III) and black hair (phototype V).

**Figure 2 ijms-22-04465-f002:**
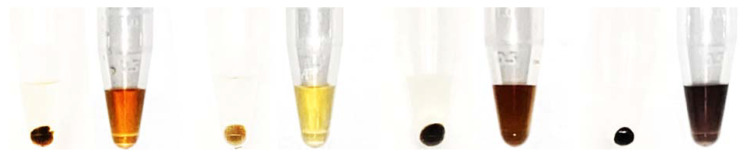
Images of isolated melanins showing two fractions: spin-down melanosomes (**left**-hand tubes) and supernatants of melanin nanoaggregates (**right**-hand tubes).

**Figure 3 ijms-22-04465-f003:**
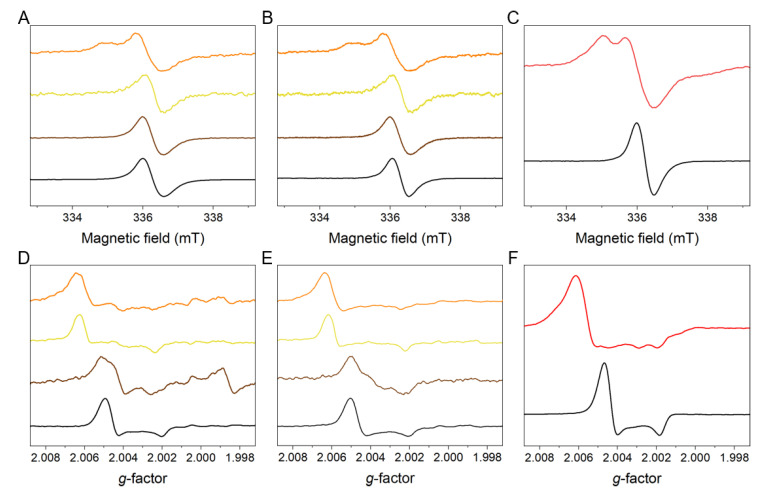
EPR spectra of the studied melanins obtained in X-band (**A**–**C**) and W-band (**D**–**F**). (**A**,**D**) show spectra of melanosomes, whereas (**B**,**E**) show spectra of melanin nanoaggregates. Color lines indicate the following samples: orange (red hair), yellow (blond hair), brown (chestnut hair) and black (black hair). (**C**,**F**) show spectra of synthetic melanin models used as standards: Cys-L-DOPA melanin (red) and DOPA melanin (black).

**Figure 4 ijms-22-04465-f004:**
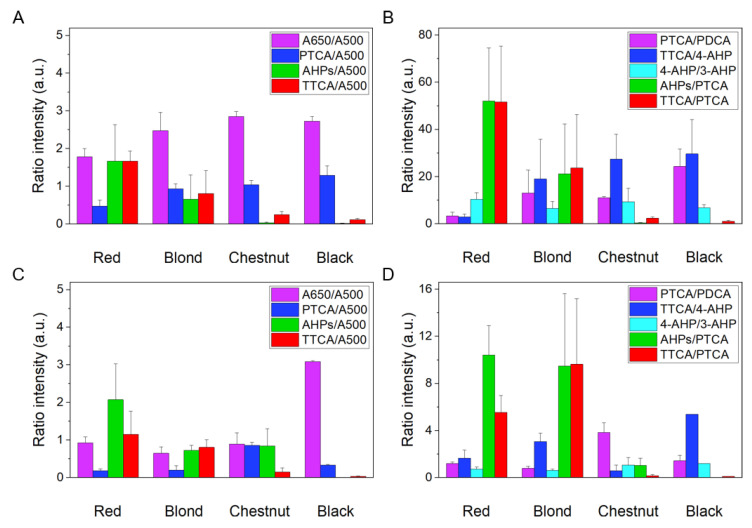
Ratios of the examined markers for melanosomes (**A**,**B**) and melanin nanoaggregates (**C**,**D**) of the studied melanins. Values represent mean ± SEM. 4-AHP/3-AHP, AHPs/PTCA, and TTCA/PTCA ratios in melanosomes are multiplied by a factor of 10 to increase clarity. A650/A500 ratios are also multiplied by a factor of 10.

**Figure 5 ijms-22-04465-f005:**
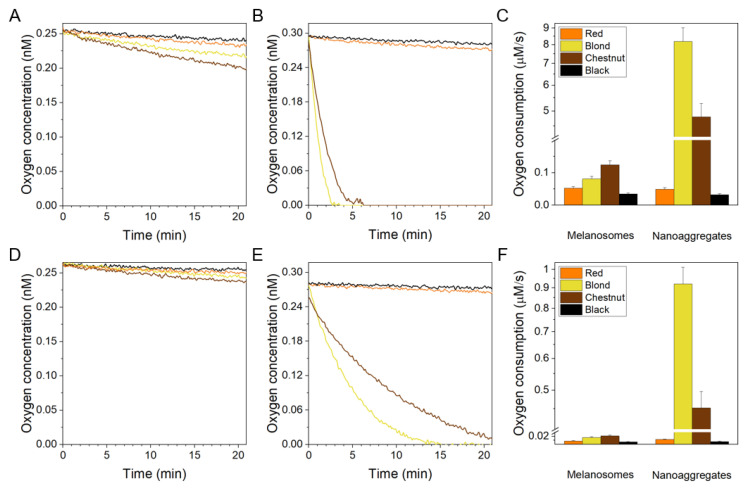
Photoinduced consumption of oxygen by natural melanin pigments under 365 nm (**A**–**C**) and 445 nm (**D**–**F**) irradiation. (**A**,**D**) show results for melanosomes whereas (**B**,**E**) show results for melanin nanoaggregates. (**C**,**F**) show histograms of initial intensities of oxygen consumption for both melanosomes and nanoaggregates of different hair melanins. Colors lines indicate the following samples: orange (red hair), yellow (blond hair), brown (chestnut hair), and black (black hair).

**Figure 6 ijms-22-04465-f006:**
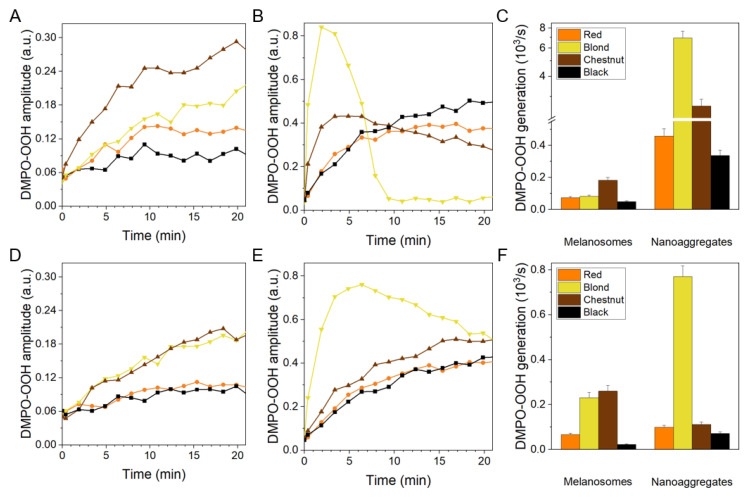
Kinetics of superoxide anion generation, detected as the formation of DMPO-OOH spin adduct during irradiation of the samples with 365 nm (**A**–**C**) and 445 nm (**D**–**F**). (**A**,**D**) show results for melanosomes whereas (**B**,**E**) show results for melanin nanoaggregates. (**C**,**F**) show histograms of initial velocities of DMPO-OOH generation by both melanosomes and nanoaggregates of different hair melanins. Color lines indicate the following samples: orange (red hair), yellow (blond hair), brown (chestnut hair), and black (black hair).

**Figure 7 ijms-22-04465-f007:**
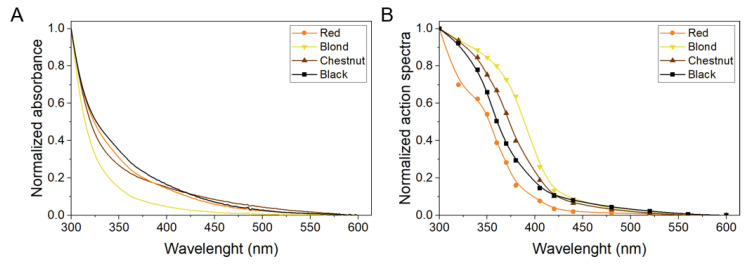
Normalized UV-Vis absorption spectra (**A**) and action spectra of singlet oxygen photogeneration (**B**) for melanin nanoaggregates from different hair samples.

**Figure 8 ijms-22-04465-f008:**
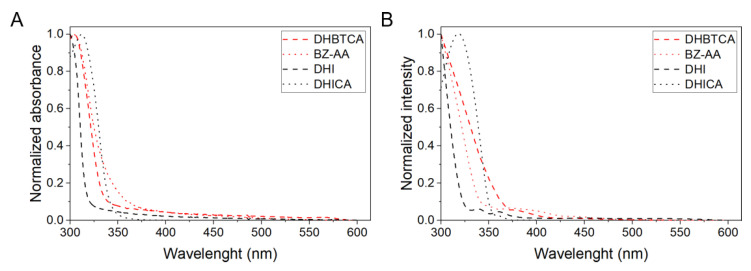
Normalized UV-Vis absorption spectra (**A**) and action spectra of singlet oxygen photogeneration (**B**) for melanin monomers.

**Figure 9 ijms-22-04465-f009:**
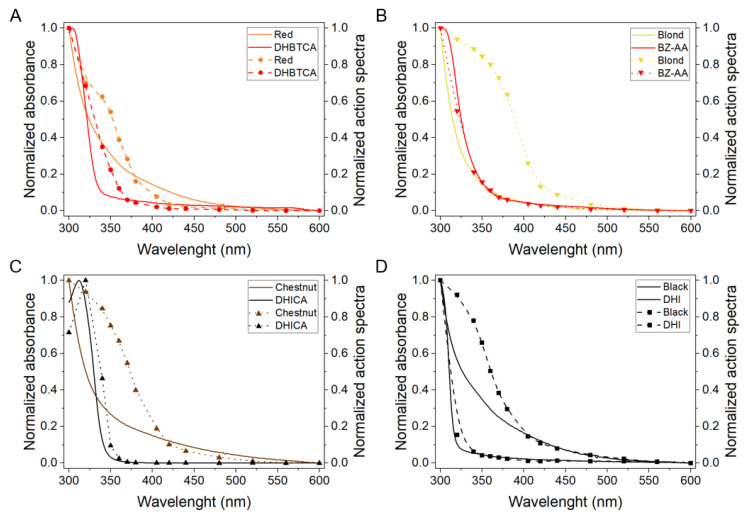
Absorption spectra of melanin nanoaggregates obtained from different hair and of selected melanin monomer units, and their corresponding action spectra of oxygen photogeneration for: red hair and BT monomers (**A**), blond hair and BZ monomers (**B**), chestnut hair and DHICA monomers (**C**), and black hair and DHI monomers (**D**). Solid lines depict absorption, whereas dashed lines with symbols characterize melanin monomer units.

**Figure 10 ijms-22-04465-f010:**
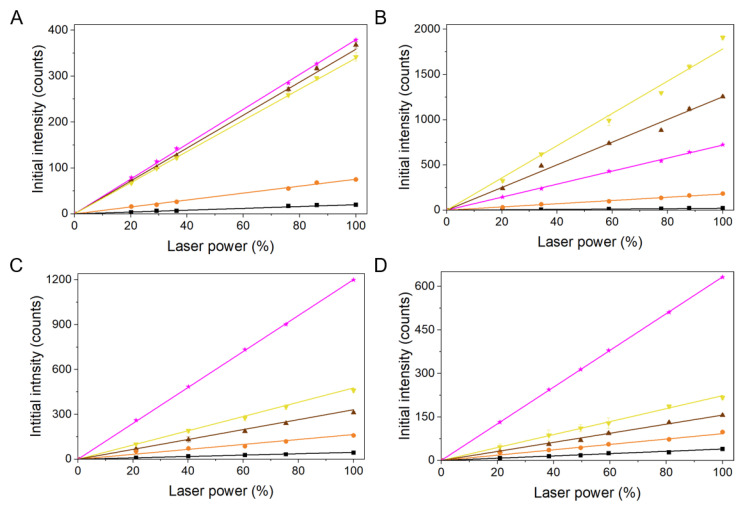
Determination of quantum yield of singlet oxygen photogeneration by melanin nanoaggregates from different hair at: 300 nm laser excitation (**A**), 332 nm laser excitation (**B**), 365 nm laser excitation (**C**), and 445 nm laser excitation (**D**). Color lines indicate the following samples: orange (red hair), yellow (blond hair), brown (chestnut hair), and black (black hair). Magenta in **A**, **B**, and **D** represents fluorescein as a reference sample, whereas magenta in **C** represents proflavine used as a standard.

**Table 1 ijms-22-04465-t001:** Quantum yields of singlet oxygen photogeneration at different wavelengths for melanin nanoaggregates from different hair and their quenching rate constants. Data represent mean ± S.D.

Type of Hair Melanin	Quantum Yield at 300 nm	Quantum Yield at 332 nm	Quantum Yield at 365 nm	Quantum Yield at 445 nm	Quenching Rate Constant [(mg/mL)^−1^s^−1^]
Red	(6.67 ± 0.53) × 10^−3^	(8.42 ± 0.69) × 10^−3^	(1.46 ± 0.09) × 10^−2^	(5.03 ± 0.27) × 10^−3^	(1.33 ± 0.12) × 10^5^
Blond	(3.02 ± 0.21) × 10^−2^	(8.38 ± 0.70) × 10^−2^	(4.22 ± 0.25) × 10^−2^	(1.18 ± 0.08) × 10^−2^	(1.12 ± 0.09) × 10^5^
Chestnut	(3.22 ± 0.23) × 10^−2^	(5.79 ± 0.39) × 10^−2^	(2.87 ± 0.20) × 10^−2^	(6.15 ± 0.41) × 10^−3^	(2.24 ± 0.18) × 10^5^
Black	(1.87 ± 0.12) × 10^−3^	(1.09 ± 0.07) × 10^−3^	(4.08 ± 0.21) × 10^−3^	(2.01 ± 0.14) × 10^−3^	(1.83 ± 0.11) × 10^5^

**Table 2 ijms-22-04465-t002:** Quantum yields of singlet oxygen photogeneration at different wavelengths for melanin subunits and their quenching rate constants. Data represent mean ± S.D.

Melanin Subunit	Quantum Yield at 300 nm	Quantum Yield at 332 nm	Quantum Yield at 365 nm	Quantum Yield at 445 nm
DHBTCA	(1.97 ± 0.14) × 10^−3^	(2.48 ± 0.16) × 10^−3^	(1.72 ± 0.09) × 10^−3^	(3.69 ± 0.18) × 10^−4^
BZ-AA	(4.65 ± 0.23) × 10^−3^	(4.92 ± 0.25) × 10^−3^	(4.07 ± 0.21) × 10^−3^	(1.01 ± 0.04) × 10^−4^
DHICA	(9.24 ± 0.56) × 10^−3^	(1.11 ± 0.07) × 10^−2^	(3.74 ± 0.19) × 10^−3^	(1.92 ± 0.15) × 10^−4^
DHI	(2.03 ± 0.12) × 10^−3^	(2.91 ± 0.23) × 10^−3^	(7.49 ± 0.40) × 10^−4^	(1.08 ± 0.04) × 10^−4^

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
