# Peer review of "Photoreactivity of Hair Melanin from Different Skin Phototypes—Contribution of Melanin Subunits to the Pigments Photoreactive Properties"

_ijms, 2021, doi:10.3390/ijms22094465_

Round 1

Reviewer 1 Report

In this study, authors performed several chemical and physical analyses of the melanina pigment in hair samples from individuals with different skin phototypes (I, II, III, and V as control). As expected a distinct photoreactivity was found into the different hair melanin types.

Although mostly confirmatory of the literature, data further underline that melanin of the individuals with the lowest phototype possesses an elevated photoreactivity, which may be in turn involved in generating reactive oxygen species in the skin of each individual with that phototype.

Experimental plan is well developed and data appropriately presented.

Author Response

We gratefully acknowledge the positive comments made by the Reviewer. 

Reviewer 2 Report

An interesting original article evaluating photoreactive properties of natural melanins from hair samples obtained from different skin phototypes. 

The article will be eligible for publication after a round of peer review :

in the material and methods section, a sub-section about statistical analysis is required.

The authors should specify what tests have been used in order to assess statistical significance, and what statistical program (and its/their maker/location) was used.

Author Response

We gratefully acknowledge the positive comments made by the Reviewer. In the revised version of our manuscript, we added a sub-section regarding statistical analysis used in the study. The analysis was made using OriginPro software. Due to the fact that the experiments were repeated 3 times (hence the n number was 3), we did not assess statistical significance analysis. 

Reviewer 3 Report

A novel article exploring photoreactivity of the various melanins derived from the hair of different phototypes; I have some queries:

I think that the title should be changed to "Photoreactivity of natural melanins of the hair from different skin phototypes – contribution of melanin subunits to the pigments photoreactive properties", as the melanin is derived from the hair shaft;

Statistical analysis should be added in material and methods.

Author Response

We gratefully acknowledge the positive comments made by the Reviewer. In the revised version of our manuscript, we modified the title, which is now the following: “Photoreactivity of hair melanin from different skin phototypes – contribution of melanin subunits to the pigments photoreactive properties”. We also added a sub-section regarding statistical analysis used in the study.